# Arg18 Substitutions Reveal the Capacity of the HIV-1 Capsid Protein for Non-Fullerene Assembly

**DOI:** 10.3390/v16071038

**Published:** 2024-06-27

**Authors:** Randall T. Schirra, Nayara F. B. dos Santos, Barbie K. Ganser-Pornillos, Owen Pornillos

**Affiliations:** 1Department of Molecular Physiology and Biological Physics, University of Virginia, Charlottesville, VA 22903, USAnayara.santos@biochem.utah.edu (N.F.B.d.S.); 2Department of Biochemistry, University of Utah, Salt Lake City, UT 84112, USA

**Keywords:** virus assembly, capsid, symmetry, vertex

## Abstract

In the fullerene cone HIV-1 capsid, the central channels of the hexameric and pentameric capsomers each contain a ring of arginine (Arg18) residues that perform essential roles in capsid assembly and function. In both the hexamer and pentamer, the Arg18 rings coordinate inositol hexakisphosphate, an assembly and stability factor for the capsid. Previously, it was shown that amino-acid substitutions of Arg18 can promote pentamer incorporation into capsid-like particles (CLPs) that spontaneously assemble in vitro under high-salt conditions. Here, we show that these Arg18 mutant CLPs contain a non-canonical pentamer conformation and distinct lattice characteristics that do not follow the fullerene geometry of retroviral capsids. The Arg18 mutant pentamers resemble the hexamer in intra-oligomeric contacts and form a unique tetramer-of-pentamers that allows for incorporation of an octahedral vertex with a cross-shaped opening in the hexagonal capsid lattice. Our findings highlight an unexpected degree of structural plasticity in HIV-1 capsid assembly.

## 1. Introduction

The HIV-1 capsid performs multiple essential functions in virus replication—it encapsulates the viral genome for delivery into host cells, and upon entry, it facilitates reverse transcription of the viral genome, protects the viral nucleic acids from innate immune sensors, engages cytoskeletal motors to transport the viral core to the nucleus, functions as a karyopherin to facilitate passage of the core through the nuclear pore complex, and eventually uncoats to release the viral replication complex that integrates into the host chromosomes [1,2,3,4,5,6,7,8]. These various capsid functions are mediated by interactions between the capsid and a growing list of host factors that bind to different sites displayed on the capsid surface. Capsid multifunctionality is further underscored by the recent clinical deployment of lenacapavir, a capsid-targeting drug that inhibits multiple steps of the HIV-1 replication cycle with up to picomolar potency [9,10].

The HIV-1 capsid is a fullerene cone made up of hexamers and pentamers of the CA protein [11]. About 200 or more CA hexamers, arranged on a hexagonal lattice, form the body of the cone. Pentamers generate sharp changes in lattice curvature—called declinations—exactly 12 of which are required to form a fully closed capsid shell. The ability of the HIV-1 CA protein to generate two different capsomers is controlled by a molecular switch—a Thr-Val-Gly-Gly (TVGG) motif—that adopts two distinct conformations which dictate whether a CA subunit will form a hexamer or pentamer [12,13]. The default state of the TVGG switch appears to be an extended, random coil configuration that enforces the hexameric state of CA; refolding of the switch into a 3_10_ helix configuration enforces the pentameric state [12].

While it is not yet established precisely what triggers CA pentamer formation as the capsid assembles in actual virions, the cellular metabolite inositol hexakisphosphate (IP_6_) induces pentamer formation when the HIV-1 CA protein assembles in vitro [12,14,15]. In contrast to NaCl, which induces the assembly of CA tubes made only of hexamers, IP_6_ induces CA assembly into fullerene cones that incorporate pentamers. Two IP_6_ molecules bind within the central channels of both the hexamer and pentamer, one above and a second below a ring of positively charged CA Arg18 sidechains. In the pentamer, but not the hexamer, the lower IP_6_ molecule is coordinated by direct contacts with a second ring of positively charged Lys25 sidechains, explaining the pentamer-specific effect of IP_6_ [12]. Correspondingly, Arg18 is essential for IP_6_-induced formation of both the hexamer and pentamer, whereas Lys25 is essential only for the pentamer (although it also contributes to efficient assembly of the hexamer [14,15]). The IP_6_ binding sites are distant from the hexamer/pentamer switch and thus must trigger refolding of the TVGG switch through allosteric cooperativity [12].

Interestingly, amino-acid substitutions of Arg18 also induce pentamer formation of HIV-1 CA in vitro. Lacking the IP_6_ binding site, Arg18 mutants are insensitive to assembly induction by IP_6_ [14] but readily assemble in 1 M NaCl. Initially, it was shown that the R18A mutation alters the phenotype of NaCl-induced CA assemblies from tubes to mixtures of spheres, cones, and capped cylinders, which are indicative of pentamer incorporation into the assembling lattice [16]. Subsequently, it was found that the R18L mutation induces CA assembly into predominantly spherical particles (as opposed to a broader mixture of shapes), which indicate even more efficient pentamer incorporation than R18A [17]. The simplest model that explains these observations is that, somehow, removal of the Arg18 sidechain has the same effect as IP_6_ binding in triggering the refolding of the TVGG motif to promote pentamers. Here, we test this model experimentally by solving and analyzing the cryoEM structures of capsid-like particles (CLPs) assembled from Arg18 mutant CA proteins in vitro. Surprisingly, we find that the Arg18 substitutions promote the formation of an alternative CA pentamer configuration that does not involve refolding of the TVGG motif. Furthermore, we find that the Arg18 mutant CLPs are polymorphic, and they incorporate octahedral vertices into the canonical fullerene geometry of the native capsid. These studies reveal unexpected plasticity in the assembly properties of the HIV-1 CA protein.

## 2. Materials and Methods

### 2.1. Purification and Assembly of HIV-1 CA Arg18 Mutants

HIV-1 CA proteins harboring the R18A, R18L, and R18G mutations were expressed in *E. coli* BL21(DE3) cells and purified as previously described [16]. CLPs were obtained by incubating purified protein (at 15 mg/mL or higher) for 1–2 h at 37 °C in high-salt assembly buffer (50 mM Tris, pH 8.0, 1 M NaCl, 5 mM β-mercaptoethanol). R18L CLPs bound to CPSF6-FG peptide (GTPVLFPGQPFGQPPLG, N-terminally acetylated and C-terminally amidated; Celltein) were obtained either by incubating pre-assembled CLPs with 3 mM peptide, or by assembling CA in low-salt buffer (50 mM Tris, pH 8.0, 0.1 M NaCl, 5 mM β-mercaptoethanol) containing 3 mM peptide.

### 2.2. CryoEM Grid Preparation and Image Data Collection

CLPs were diluted 2- to 10-fold into 0.1 M KCl, then immediately applied (3–4 μL) to glow-discharged lacey carbon 300-mesh copper grids. The grids were briefly blotted manually and plunge-frozen in liquid ethane using a home-built device. CryoEM data were collected at the University of Virginia Molecular Electron Microscopy Core. Videos were collected using a Krios (ThermoFisher, Waltham, MA, USA) operating at 300 kV and equipped with an energy filter and K3 direct detector (Gatan, Pleasanton, CA, USA) or using a Glacios (ThermoFisher) operating at 200 kV and equipped with a Falcon 4 detector (ThermoFisher). Data were collected using EPU (ThermoFisher) with a total dose of approximately 50 electrons/Å^2^ over 40 frames and target defocus of 1.0 to 2.5 μm. Raw movies were corrected for beam-induced motion using MotionCor2 [18].

### 2.3. Structure Determination

All image processing, CTF estimations, and map calculations were performed using cryoSPARC [19]. Initial particles were manually picked to generate references for subsequent template-based picking. Multiple rounds of reference-free, two-dimensional (2D) classification (performed as described [12]), were used to identify high-curvature particle sets that contained pentamers. Particle alignments were performed as described [12]. In brief, the particles were initially extracted in large box sizes and binned, then aligned (C1 symmetry) to obtain initial maps. The maps were oriented and centered to focus on regions of interest (e.g., hexamer or pentamer) and then refined with a high-pass filter (20 Å) in C1 symmetry. After discarding overlaps, the particles were then re-extracted in small box sizes, unbinned, and further refined locally with applied symmetry as indicated. Details on the numbers of images collected, numbers of particles extracted, and map calculation flowcharts are provided in Appendix A and Table 1.

To avoid confusion with CLPs, from this point forward, the term “particle” exclusively refers to the boxed units that are picked from the cryomicrographs and used in image reconstruction; depending on the box size, one or more particles can be picked from each CLP.

### 2.4. Coordinate Modeling

The coordinate model for the R18L hexamer was built by docking a WT hexamer subunit (PDB 8ckv [13]) into the map. Secondary structure, non-crystallographic symmetry, and reference model restraints were used in iterative rounds of model building using Coot [20] and real-space refinement using the Phenix suite [21]. The coordinate model for the R18L pentamer was built by docking the final refined R18L hexamer subunit into each of the symmetry-distinct five pentamer subunits, followed by rigid-body refinements, treating the N-terminal domains (NTD) and C-terminal domains (CTD) as separate units. The coordinate model statistics are given in Table 2.

Structural depictions for figures were made using Chimera and PyMol (Schrödinger Scientific).

## 3. Results

### 3.1. CryoEM Structures of R18L HIV-1 CA Assemblies

We initially analyzed R18L CLPs, which were assembled by incubating the mutant HIV-1 CA protein in 1 M NaCl [17]. In contrast to WT CA, which forms tubes under the same conditions, the R18L CA assemblies are more capsid-like in morphology. The R18L CLPs are generally smaller and more rounded compared to canonical fullerene cones, but still display variation in shape (Figure 1A). The smallest, roundest CLPs are about the size of T = 3 or T = 4 icosahedral capsids (~35 nm diameter) [22,23]. However, reference-free 2D class averages of these CLPs show irregular density distributions in the capsid walls (Figure 1B); such irregularities are indicative of geometric defects and a lack of global symmetry [24]. Thus, rather than reconstructing the entire CLP structure, we used recently described single-particle averaging strategies to focus on the locally symmetric building blocks [12,13] (Appendix A). CryoEM maps centered on the R18L hexamer and pentamer (Figure 1C,D) were refined to nominal resolutions of 3.7 Å and 4.4 Å, respectively (Appendix A).

A comparison with published structures indicates that the R18L CA hexamer is very similar to the WT CA hexamer found within native capsids and in vitro assembled fullerene cones (Appendix A). In contrast, the R18L pentamer unexpectedly adopts a configuration that is distinct from the WT pentamer. Instead, the R18L pentamer is similar to the previously described crystal structure of a disulfide-stabilized HIV-1 CA pentamer harboring the N21C and A22C mutations [25]. The correspondence between the R18L and N21C/A22C pentamers (which we call here the “alternative” form) and their divergence from the WT pentamer (the “canonical” form) are illustrated in Figure 2.

### 3.2. R18L Allows Pentamer Formation without Triggering the CA Hexamer/Pentamer Switch

The HIV-1 CA hexamer and pentamer are quasi-equivalent: the NTDs use the so-called NTD–NTD interface to form a central ring that interacts with the CTDs, each of which packs against the NTD from the neighboring subunit via the NTD–CTD interface. In actual viral capsids [26] and in fullerene cones assembled in vitro [12,13], the canonical hexameric and pentameric NTD–NTD and NTD–CTD interfaces are distinct, and their configurations are dictated by the folding configuration of the TVGG switch (dashed box in Figure 3A). As previously described in detail (2.5 Å resolution), the CA N21C/A22C pentamer nevertheless has hexamer-like NTD–NTD and NTD–CTD interfaces [25]. Thus, the alternative CA pentamer forms without refolding of the TVGG motif.

Although our R18L pentamer map is at a more limited resolution (4.4 Å), the TVGG backbone density is clearly more consistent with the hexamer-state rather than the pentamer-state configuration (compare Figure 3B,C). We also observe two additional features that further support a lack of refolding of the TVGG switch in R18L (Figure 3D). Firstly, the random coil configuration of the TVGG motif is normally stabilized by an extensive network of hydrogen bonds that includes the Arg173 sidechain from the interacting CTD. Arg173 is clearly resolved in our cryoEM map of the R18L pentamer as pointing towards the bottom of NTD helix 3, as also seen in all HIV-1 CA hexamer structures solved to date. Secondly, the Met66 sidechain, which functions as a gate, is resolved in the “closed” configuration, again as seen in the hexamer. Thus, we conclude that, like the N21C/A22C disulfide crosslink, the R18L mutation allows for assembly of the alternative pentameric CA ring, which has no need for refolding of the TVGG switch into its 3_10_ helix configuration.

Structural comparisons suggest to us that R18L bypasses the hexamer/pentamer switch by allowing closer packing of the N-terminal ends of helix 1 compared to that found in the canonical pentamer (as illustrated by the dashed circles in Figure 2C). In the R18L pentamer, the leucine sidechains are within van der Waals packing distance (Figure 1F). Similarly, previous analyses have shown that the N21C/A22C disulfide crosslink enforces close packing at this region, even with native Arg18 present [25]. Indeed, disulfide-enforced pentamer formation is even more efficient when combined with the R18L mutation [22,25]. In the canonical pentamer, the two bound IP_6_ molecules dictate the packing geometry of the NTD subunits: the top IP_6_ molecule sterically constrains packing distances of the Arg18 sidechains, and helix 1 adopts a different tilt so that the Lys25 sidechains can coordinate with the bottom IP_6_ molecule [12]. Put another way, construction of a pentameric ring while preserving the hexameric NTD–NTD and NTD–CTD interfaces is normally unfavorable because of steric constraints and the need to accommodate IP_6_; these are eliminated by the R18L mutation or overcome by disulfide crosslinking.

### 3.3. The NTD–CTD Interfaces of the R18L Pentamer and Hexamer Are Functionally Equivalent

The differing configurations of the TVGG motif in the WT CA hexamer and canonical pentamer also determine the capacity of the differing NTD–CTD interfaces to bind ligands that contain phenylalanine–glycine (FG) motifs; only the hexamer but not the pentamer is configured to bind FG [12,13,27]. Since the alternative R18L pentamer contains a hexamer-like TVGG configuration and NTD–CTD interface, we predicted that it should be also capable of binding FG ligands. To test this, we incubated the R18L CLPs with a representative FG-containing peptide (derived from the protein CPSF6) [28,29] and performed focused refinement on a map that included both hexamers and pentamers, achieving a nominal resolution of 5.4 Å (Appendix A). We found that all of the NTD–CTD interfaces in the R18L pentamer and hexamer are occupied by the FG peptide (colored green in Figure 4A,B), and that the pentamer-bound peptides have the same configuration as the hexamer-bound peptides (to the limit of the map resolution). Consistent with these results, the disulfide-crosslinked N21C/A22C pentamer has been shown to bind the small molecule inhibitors, PF74 and GS-CA1/GS-6207/lenacapavir, which compete with FG-containing ligands for the same binding site [30,31]. Thus, the NTD–CTD interface of the alternative pentamer form of HIV-1 CA shares the structure–function correlates of the hexamer but not the canonical pentamer.

### 3.4. R18L CLPs Combine Icosahedral and Octahedral Geometries

Another unexpected feature of the R18L CLPs is that the pentamers tend not to form a standard declination, which consists of a pentamer surrounded by five hexamers. Masked 2D class averages revealed pentamers connected to each other (black arrows in Figure 5A), which, in classical quasi-equivalent icosahedral systems, is only allowed in T = 1 capsids. Surprisingly, we also obtained several 2D classes which indicated that these adjacent pentamers are further organized as a tetramer-of-pentamers surrounding a cross-shaped hole (orange arrowheads in Figure 5A). A raw cryoimage of a CLP oriented with evident 4-fold symmetry is shown in Figure 5B. To examine these larger capsomer arrangements, we reconstructed cryoEM maps with larger box sizes (Appendix A). Self-rotation functions (calculated in reciprocal space [32]) clearly indicated the presence of a 4-fold rotational axis (Appendix A), and indeed, a map in C1 symmetry (Figure 5C, Appendix A) refined to lower nominal resolution than a map with imposed C4 symmetry (Figure 5D, Appendix A). The tetramer-of-pentamers (magenta/yellow in Figure 5C,D) is surrounded by eight hexamers (blue/orange in Figure 5C,D), and thus is well accommodated within the hexagonal capsid lattice. As in the fullerene cone, the R18L hexamers and pentamers are connected to each other by the CTD, which forms dimeric interactions via helix 9 and trimeric interactions via helix 10. Without exception, each CTD in our structures is engaged in the dimer interface. Four CTDs—one from each pentamer—do not engage in the trimer interface and instead surround the cross-shaped hole (Figure 5E).

The canonical CA pentamer in a fullerene capsid is completely surrounded by hexamers, with only rare instances of two pentamers directly adjacent to one another [26]. Interestingly, we could not readily identify a 2D class in the R18L datasets that unambiguously showed a single pentamer surrounded by five hexamers. To determine if there may be instances of R18L pentamers in such an arrangement, we performed extensive 2D and 3D classifications, starting with a curated set of 335,592 particles (details provided in Appendix A). We identified 183,070 particles of the tetramer-of-pentamers and 18,643 particles of the single-pentamer type; these results indicate that single-pentamer declinations do occur in the R18L CLPs, but at lower frequencies than the tetramer-of-pentamers (Figure 6A). Importantly, mapping the final refined positions of individual particles to the original micrographs indicates that both the tetramer-of-pentamers and single-pentamer classes can be derived from the same CLPs (Figure 6B). These results suggest that the R18L CLPs are not purely fullerene structures but are of a mixed symmetry—they incorporate both icosahedral and octahedral symmetries.

### 3.5. CA R18A and R18G Form the Same Tetramer-of-Pentamers as R18L

The R18A and R18G CA mutants have been characterized more extensively in the literature in the context of actual virions [33,34,35]. Virions harboring these mutations contain aberrantly assembled capsids, but fullerene cones that are apparently normal can also be observed. Thus, we also analyzed CLPs derived from the CA R18A and R18G mutants (Appendix A). The R18A CLPs were larger and had more varied shapes compared to R18L (Figure 7A), whereas R18G contained more capped cylinders and tubes (Figure 7B). In each case, focused refinement on the most highly curved regions of the CLPs also produced cryoEM maps (6.0 Å resolution for R18A and 7.1 Å for R18G) containing the alternative pentamer form in the tetramer-of-pentamers arrangement (Figure 7C,D). Although the R18L structures suggest that the leucine substitution engages in stabilizing hydrophobic contacts in the mutant pentamer (Figure 1F), the above results indicate that removal of the arginine sidechain is the predominant driver of the alternative pentamer form. We did not search for the single-pentamer arrangement in these CLPs due to limited particle numbers, but it is reasonable to assume that they are also present.

### 3.6. FG Binding Reinforces Octahedral Symmetry in Arg18 Mutant CA

Since the alternative pentamer binds to and is stabilized by FG-containing ligands, we wondered if FG binding would reinforce the tetramer-of-pentamers and thereby induce purely octahedral assemblies. To test this notion, we assembled R18L CA in the presence of excess CPSF6-FG peptide in buffer containing only 0.1 M NaCl (Figure 8A). In this low-salt condition, assembly is FG-dependent because CLPs did not form in the absence of the peptide. Comparing the FG/low-salt to the no-FG/high-salt assemblies, the CLPs assembled with FG had smaller sizes, narrower size distributions, and higher sphericities (Figure 8B). Furthermore, a good fraction of the 2D class averages showed CLPs with relatively even density distributions for the capsid walls, indicating that these CLPs have fewer “defects” and are more globally symmetric (compare Figure 8C with Figure 1B). Indeed, about 50% of the selected particles produced an ab initio map that was almost completely spherical, even though the map was calculated in C1 (without an initial template and without imposing symmetry) (boxed in red in Appendix A). Self-rotation functions indicated the presence of three mutually orthogonal 4-fold rotation axes (Appendix A); thus, the initial map was refined with octahedral (O) symmetry imposed to 4.0 Å nominal resolution (Figure 8D, Appendix A). The asymmetric unit consists of one pentamer, one-half of a hexamer, and one-third of a hexamer (Figure 8E). The tetramer-of-pentamers surrounds each of the octahedral 4-fold symmetry axes, whereas the (pseudo) 6-folds of the hexamers coincide with the octahedral 2-fold and 3-fold symmetry axes. All unique NTD–CTD interfaces in the asymmetric unit contained bound FG ligand (Figure 8D). Altogether, these data indicate that, indeed, the FG ligand reinforces octahedral geometry in the R18L assemblies.

## 4. Discussion

The fullerene cone is a flexible extension of the Caspar–Klug quasi-equivalence theory of virus capsid architecture based on icosahedral geometry [36]. A fully closed cone contains exactly twelve pentavalent positions (occupied by CA pentamers) and variable numbers of hexavalent positions (occupied by CA hexamers). The pentamers and hexamers are held together by distinct NTD–NTD and NTD–CTD interfaces, and they connect to each other through 2-fold symmetric (CTD dimer) and 3-fold symmetric (CTD trimer) inter-capsomer bonding interfaces. The locally symmetric CA–CA interactions—6-fold, 5-fold, 3-fold, and 2-fold—correspond to the allowed symmetry axes in quasi-equivalent icosahedral shells. Unexpectedly, we find here that HIV-1 CA harboring Arg18 mutations can perturb the capsid structure in two ways. Firstly, the pentavalent CA–CA interactions pack in an alternative fashion, utilizing similar NTD–NTD and NTD–CTD interfaces as the hexamer. An important property of this alternative pentamer form is that it shares the ability of the hexamer to bind FG motif-containing ligands. Secondly, the alternative pentamer form of CA can be further organized as a tetramer-of-pentamers, allowing for the incorporation of a 4-fold symmetric (octahedral) vertex within the fullerene assembly. The inter-capsomer contacts in this vertex are similar to those in the native fullerene cone, except that each of the CTD subunits surrounding the 4-fold has an unsatisfied binding surface. Interestingly, the nonbonded CTD element is helix 10, which normally makes the CTD trimer interactions in fullerene cones; this is the most variable and “slippery” of the interfaces in native capsids [13].

Naturally occurring octahedral viral capsids have not been identified, but the murine polyoma virus capsid protein, called VP1, can assemble into an octahedral shell in vitro [37]. Depending on buffer conditions, VP1 assembles into discrete CLPs of either octahedral or icosahedral geometry. Like VP1, our mutant CA can also form a discrete octahedral shell, but unlike VP1, CA has greater flexibility and integrates the tetramer-of-pentamers within fullerene-like shells, thereby combining icosahedral and octahedral symmetries in the same CLP. We have not found examples of such capsids in the literature, although this phenomenon might explain some of the previously described symmetry-breaking “defects” in virus capsids [24].

Can the alternative pentamer exist in the native HIV-1 capsid? One possibility is that the 4-fold vertex may be stochastically incorporated into capsids as they assemble in situ. Such a vertex has not been reported in extensive lattice mappings of HIV-1 and retroviral cores performed using sub-tomogram averaging [1,26,38,39]. Sub-tomogram lattice mapping is a template-based approach; therefore, it is possible that rare instances of this arrangement may have been missed in prior studies. However, IP_6_ induces and stabilizes the canonical HIV-1 CA pentamer by coordinating the native Arg18 ring [12,27]. We believe that the mM-level concentrations of IP_6_ inside virions [40] would disfavor the alternative pentamer configuration during native capsid assembly.

A second possibility is that the 4-fold vertex may form post-assembly through local rearrangement of the CA subunits. We considered theoretical mechanisms by which the tetramer-of-pentamers might be derived from the canonical geometries of a fullerene cone. With the constraint of invoking minimal changes in quaternary bonding interactions, the simplest model that we could envision (Figure 9A) is to start with two hexamers and two pentamers in a local T = 3 arrangement. Removing the two CA subunits that connect the two hexamers and reclosing these rings into pentamers would produce the tetramer-of-pentamers. Interestingly, such a change may only require perturbing the four central capsomers and not the eight encircling hexamers. Still, this would involve extensive changes in inter-subunit packing: breaking four NTD–NTD, four NTD–CTD, and two CTD trimer interfaces (albeit these are already distorted and weak in this highly curved region [13]), as well as reclosing the two former hexamer subunits into pentamers and rearranging the two former canonical pentamers into the alternative configuration. The bound IP_6_ may also need to dissociate from the canonical hexamers and pentamers as they rearrange into the tetramer-of-pentamers. Our crude modeling suggests that local lattice curvature might only change minimally (Figure 9B), but this needs to be confirmed more rigorously because the R18L octahedron has the same number of CA subunits and is more similar in size to a T = 4 icosahedron (240 subunits) than to T = 3 (180 subunits). We are not aware of an obvious molecular process that would induce such a change in actual HIV-1 capsids, but we note that so-called “capsid remodeling”—or rearrangement of the CA lattice—has been suggested to occur in the post-entry stage of HIV-1 replication, particularly as the capsid passes through the nuclear pore [41,42]. The nuclear pore channel is a meshwork of FG-containing nucleoporins, with an estimated FG concentration of 50 mM [43,44]. Given our finding that excess FG ligand promotes octahedral assembly of R18L CA in vitro, we speculate that a similar, FG-induced pentamer remodeling might occur during HIV-1 nuclear entry.

A third possibility is that the octahedral vertex is only an in vitro phenomenon that is unique to Arg18-mutant CA that the native protein does not access biologically. In the case of polyoma virus VP1, the in vitro octahedron has no known biological function as yet, but these assemblies have garnered considerable attention as nanocages, with potential applications in medicine and materials science (e.g., [45,46]). Because the HIV-1 capsid is a karyopherin [6,7], HIV-1 CA assemblies offer the potential of delivering cargo directly inside target cell nuclei, which could be advantageous in terminally differentiated cells. Taken together with other studies [12,17,22,23,27], our work here supports the possibility that nanocages based on HIV-1 CA may be engineered to tailor sizes, symmetries, and surface functionalities for specific cargo or delivery requirements.

Our finding that a single amino acid mutation is sufficient to trigger octahedral packing suggests to us that genetic barriers to alternative assemblies by retroviral CA proteins may not be insurmountably high. It is tempting to speculate that some hitherto understudied retrovirus may have evolved to incorporate octahedral symmetry into its native capsid.

Previous studies have indicated that the principal defect in Arg18-mutant HIV-1 virions manifests when the conical capsid assembles during the maturation step of virus replication [33]. Our studies now reveal in molecular detail that the assembly defect is due to the induction of an alternative CA pentamer. Interestingly, a sub-population of R18A and R18G mutant virions can contain apparently normal, cone-shaped capsids [33,34]. Post-assembly, the Arg18 ring in the hexamer has been shown to facilitate uptake of deoxynucleotide triphosphates (dNTPs), which are required for reverse transcription but are concentration-limited in non-dividing cells [35]. Thus, the lack of infectivity of Arg18 mutant virions can be explained by a two-stage mechanism: the majority of virions do not assemble capsids properly, and those that still manage to form apparently normal capsids are still non-infectious because they cannot uptake dNTPs. However, it can be argued that the latter sub-population of Arg18 mutant virions should gain some infectivity in actively dividing cell lines wherein dNTP concentrations are not limiting, and yet this is not the case [33,34]. We suggest that this observation indicates the requirement for some pentamer-specific function(s) in the post-entry pathway of the capsid, conferred by the canonical but not the alternative pentamer form that we observe here. Further studies are needed to explore this possibility.

## Figures and Tables

**Figure 1 viruses-16-01038-f001:**
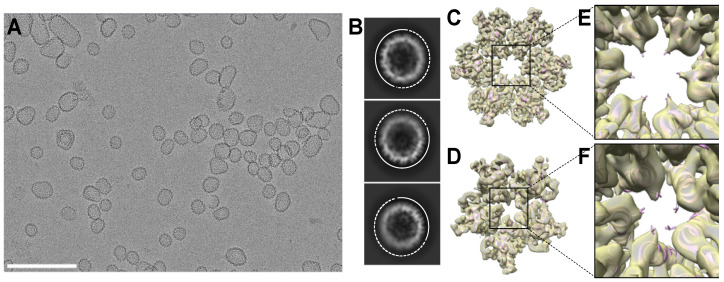
Assembly and structure of R18L capsid-like assemblies. (**A**) Representative cryomicrograph of R18L CLPs assembled by incubating the protein in 50 mM Tris, pH 8.0, 1 M NaCl, and 10 mM β-mercaptoethanol for 1 h at 37 °C. Scale bar, 200 nm. (**B**) Examples of 2D class averages of ~35 nm CLPs. Solid and dashed arcs indicate relatively well-defined and irregular capsid walls, respectively. (**C**) Focused cryoEM reconstruction of the R18L hexamer (C6 symmetry). (**D**) Focused reconstruction of the R18L pentamer (C1 symmetry). (**E**,**F**) Close-ups of the R18L substitutions. The leucine sidechains (shown as sticks) are in van der Waals contact in the pentamer and are less solvent accessible in the pentamer than in the hexamer.

**Figure 2 viruses-16-01038-f002:**
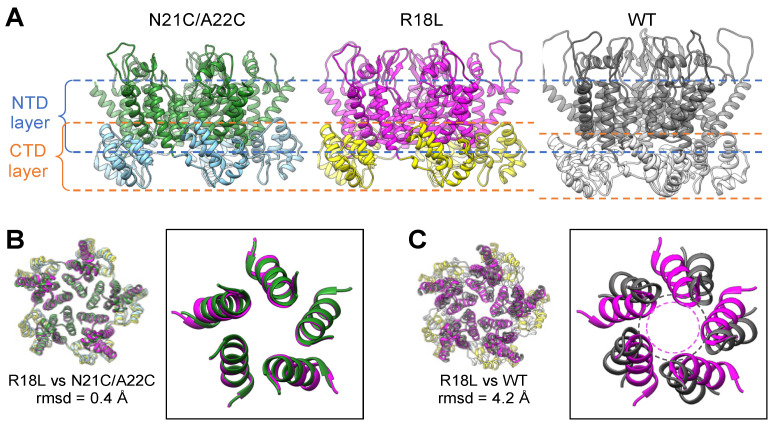
Comparison of R18L, N21C/A22C, and WT CA pentamers. (**A**) Side views of HIV-1 CA pentamers from the N21C/A22C crystal structure, PDB 3p05 [25]; the R18L structure solved in the current study; and the WT pentamer, PDB 7urn [12]. Dashed lines indicate the NTD and CTD layers. (**B**,**C**) Superposition of the indicated pentamers; rmsds are calculated for helix 1 Cα atoms after superimposing whole pentamers. Boxes show only helix 1 for clarity.

**Figure 3 viruses-16-01038-f003:**
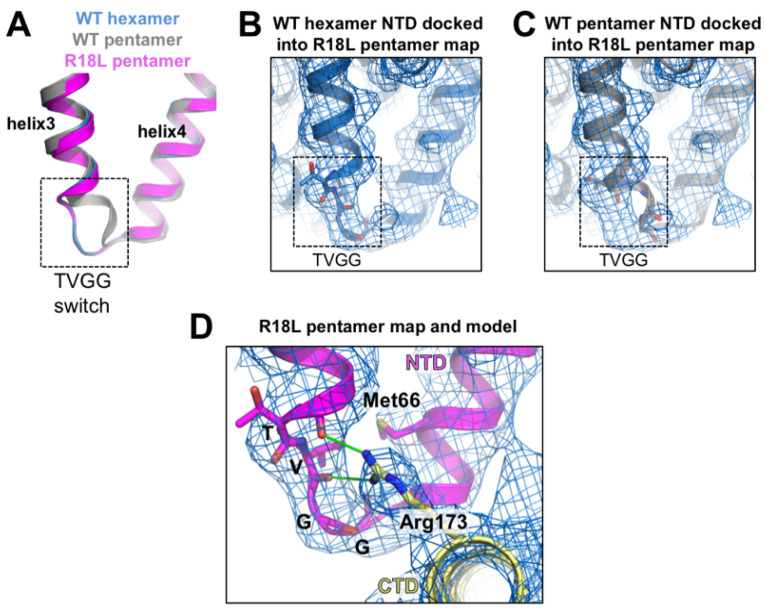
Status of the TVGG hexamer/pentamer switch. (**A**) Superposition of the R18L pentamer subunit (magenta) with the WT hexamer subunit (blue) and the WT pentamer subunit (gray) illustrating the two different configurations of the TVGG motif (inside dashed box). (**B**,**C**) The WT hexamer (**B**) or WT pentamer (**C**) (both from PDB 7urn [12]) were docked into the R18L pentamer map (blue mesh). (**D**) Details of the NTD–CTD interactions involving the TVGG motif in the R18L pentamer. The NTD is in magenta, and the CTD is in yellow. Relevant sidechains (discussed in the text) are shown as sticks. Hydrogen bonds are shown in green.

**Figure 4 viruses-16-01038-f004:**
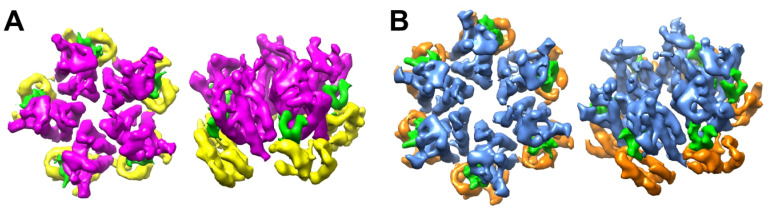
Structures of R18L capsomers in complex with FG peptide derived from the CPSF6 protein. The bound peptides are colored in green. (**A**) R18L pentamer, with NTD in magenta and CTD in yellow. (**B**) R18L hexamer, with NTD in blue and CTD in orange.

**Figure 5 viruses-16-01038-f005:**
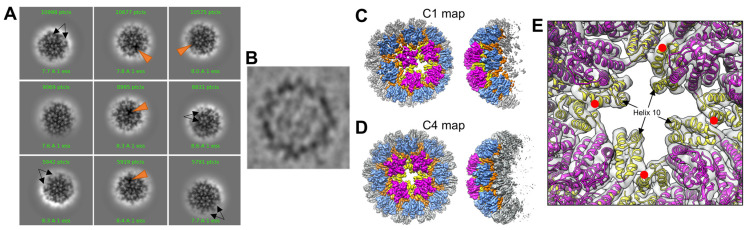
Reconstruction of the R18L tetramer-of-pentamers. (**A**) Examples of 2D class averages, with the CLP walls masked to emphasize facets. Black arrows indicate adjacent pentamers. Orange arrowheads indicate the cross-shaped hole surrounded by four pentamers. (**B**) CLP fortuitously oriented with evident 4-fold symmetry. (**C**) Map calculated in C1 (without imposed symmetry). (**D**) Map calculated with imposed C4 symmetry. The pentamers are colored in magenta (NTD) and yellow (CTD); surrounding hexamers are colored in blue (NTD) and orange (CTD). (**E**) Detail on the CTDs surrounding the cross-shaped hole. Red dots indicate the CTD–dimer interface that connects adjacent pentamers. Helix 10 surrounds the cross-shaped hole.

**Figure 6 viruses-16-01038-f006:**
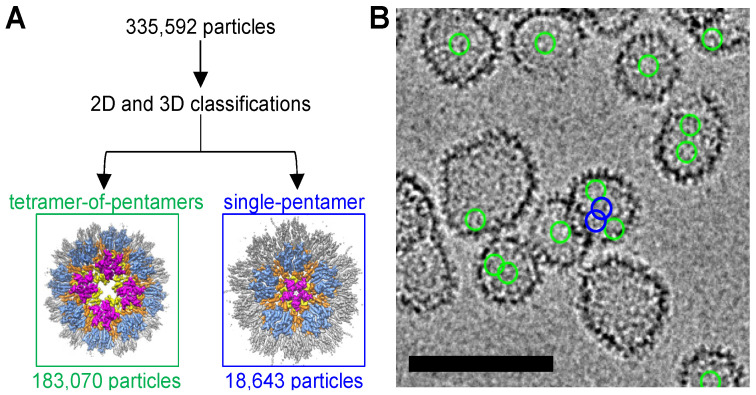
R18L CLPs contain two types of pentamer arrangements. (**A**) Classification scheme to identify and segregate particles belonging to the tetramer-of-pentamer and single-pentamer types of vertexes. (**B**) Representative micrograph indicating the final refined positions of the tetramer-of-pentamers (green circles) and single pentamer (blue circles). Scale bar, 100 nm.

**Figure 7 viruses-16-01038-f007:**
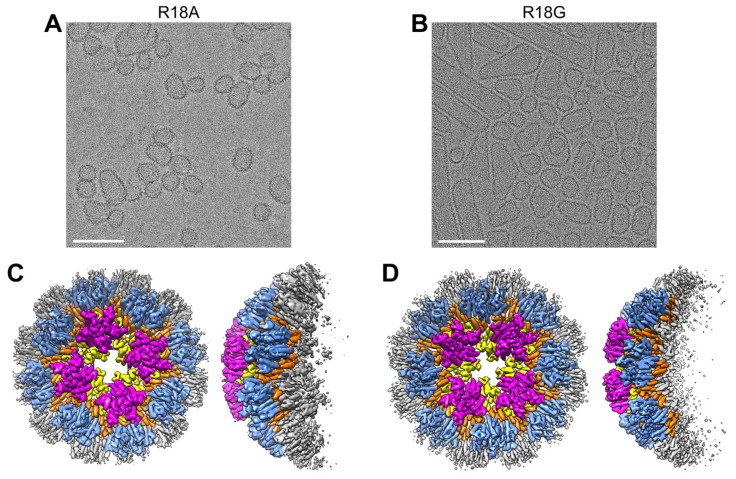
Structures of the tetramer-of-pentamers in other Arg18 mutant CA CLPs. (**A**,**B**) CLPs of the indicated mutants. Scale bars, 100 nm. (**C**,**D**) Corresponding reconstructions with imposed C4 symmetry.

**Figure 8 viruses-16-01038-f008:**
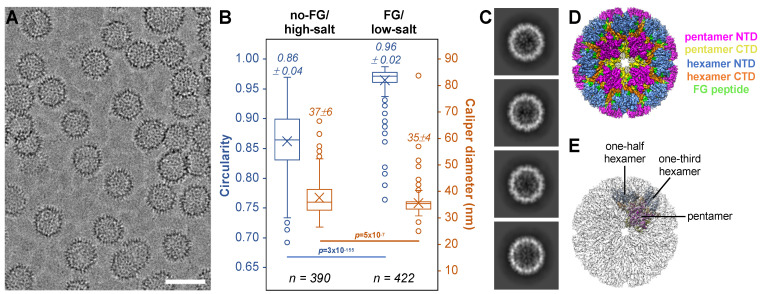
FG-induced assembly enhances the octahedral character in R18L CLPs. (**A**) CLPs formed by R18L when assembled in low-salt buffer and 3 mM CPSF6-FG peptide. Scale bar, 50 nm. (**B**) Dimensions and shape of R18L CLPs induced by either 1 M NaCl (no-FG/high-salt) or 3 mM FG peptide (FG/low-salt). Several representative high-defocus cryomicrographs were selected at random from the datasets used for structure determination. Measurements were performed on all quantifiable CLPs found in each image (criteria: ice-embedded and not on carbon, defined walls, minimal overlap). A circularity of 1 indicates a perfectly spherical capsid. The caliper diameter indicates the longest dimension. Data are shown as box-and-whisker plots, where horizontal lines demarcate the quartiles, × indicates the mean, and open circles indicate outliers; *n* indicates the number of CLPs measured; *p* values are determined based on two-tailed t tests, assuming unequal variances. (**C**) Examples of 2D classes showing more uniformly defined CLP wall densities. (**D**) CryoEM structure of FG-induced R18L octahedron. (**E**) Translucent map with the octahedral asymmetric unit, which consists of a single pentamer, one-half of a hexamer, and one-third of a hexamer.

**Figure 9 viruses-16-01038-f009:**
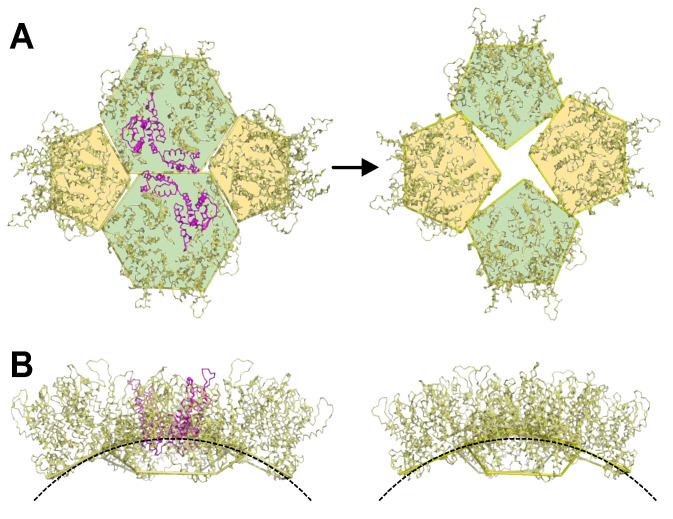
Speculative mechanism of local “remodeling” of CA subunits into a tetramer-of-pentamers. (**A**) Starting with two hexamers and two pentamers with a local T = 3 arrangement, removing two hexamer subunits (magenta), reclosing the former hexamers into pentamers, and rearranging the former canonical pentamers into alternative form generates a tetramer-of-pentamers. (**B**) Side views comparing the local curvatures.

**Table 1 viruses-16-01038-t001:** Image data collection and processing statistics.

	R18L	R18L	R18L	R18L	R18A	R18G	R18L High-Salt + FG	R18L Low-Salt + FG	R18L Low-Salt + FG
EMDB	EMD-43641	EMD-43642	EMD-43632	EMD-43633	EMD-43634	EMD-43635	EMD-43636	EMD-43637	EMD-43638
Focus region	Hexamer	Pentamer	Tetramer of pentamers	Single pentamer	Tetramer of pentamers	Tetramer of pentamers	Tetramer of pentamers	Tetramer of pentamers	Octahed-ron
Magnification	64,000×	64,000×	53,000×	53,000×	81,000×	36,000×	36,000×	81,000×	81,000×
Voltage (kV)	300	300	300	300	300	200	200	300	300
Exposure (e^−^/Å^2^)	50	50	55	55	50	50	45	50	50
Defocus range (μm)	1.0 to 2.5	1.0 to 2.5	1.0 to 2.5	1.0 to 2.5	1.0 to 2.5	1.0 to 2.5	1.0 to 2.5	1.0 to 2.5	1.0 to 2.5
Pixel size (Å)	0.686 *	0.686 *	1.62	1.62	1.08	1.20	1.20	1.08	1.08
Symmetry imposed	C6	C1	C4	C5	C4	C4	C4	C4	O
Particles	737,346	282,138	106,564	18,643	43,954	73,987	172,654	109,290	130,325
Map resolution **	3.7	4.4	5.8	6.6	6.0	7.1	5.3	5.4	4.0
Map resolution range	2.3–5.4	3.0–7.7	3.5–75	3.9–78	4.7–67	5.2–64	3.2–44	3.4–13	2.4–8.8

* Collected in superresolution mode, standard pixel size is 1.372, ** At 0.143 FSC threshold.

**Table 2 viruses-16-01038-t002:** Coordinate modeling and refinement statistics.

	R18L Hexamer	R18L Pentamer
EMDB	EMD-43641	EMD-43642
PDB	8vxv	8vxw
Model resolution at FSC 0.143/0.5 threshold (Å) *	3.6/4.1	4.3/6.0
Map-sharpening B-factor (Å^2^)	250	272
No. of nonhydrogen atoms	1524	5020
No. of protein residues	201	1015
Average B-factors (Å^2^)	106.4	150.4
Rmsd bond lengths (Å)	0.004	0.002
Rmsd bond angles (°)	0.664	0.583
MolProbity score	1.40	0.55
Clash score	6.97	0.14
Poor rotamers (%)	0	0
Ramachandran favored (%)	98.0	99.2
Ramachandran outliers (%)	0	0
Ramachandran *Z*-score	1.04	2.40

## Data Availability

CryoEM maps and PDB models are deposited with accession numbers indicated in Table 1 and Table 2.

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
