# Peer review of "Arg18 Substitutions Reveal the Capacity of the HIV-1 Capsid Protein for Non-Fullerene Assembly"

_viruses, 2024, doi:10.3390/v16071038_

Round 1
Reviewer 1 Report
Comments and Suggestions for Authors
In this paper, the authors have structurally characterized an interesting mutant of the HIV-1 capsid protein that forms a non-canonical pentamer. This non-canonical arrangement leads to an unusual capsid formation with a different shape distribution than normal and an unusual octahedral geometry which results in a cross-shaped hole in the capsids along with other unique structural properties. The structural analysis is well-designed and well-explained. There are only a couple of minor issues to be addressed before the manuscript is suitable for publication:
1. The authors perform a nice structural comparison between the R18L capsid and previously-characterized R18A and R18G capsids. The R18 mutant capsids share some properties, but there are structural differences, particularly in terms of the distribution of oligomeric assemblies observed. The R18A and R18G have been characterized more thoroughly, so one additional thing that would tie the results together nicely would be to perform some characterization of R18L along the lines of what has been previously done for R18A and R18G. For example, the R18A and R18G mutants exhibit defective HIV particle formation and produce non-infectious particles. In addition, R18G exhibits enhanced stability in the absence of DTT relative to wild-type. Inclusion of these kinds or similar in vitro data would help tie the observed structural differences to observed biochemical differences.
2. Check the references. Reference 16 on line 68 should actually be reference 15. I didn't check all of them, but if I found one, there might be other instances where something is mis-referenced.
Author Response
In this paper, the authors have structurally characterized an interesting mutant of the HIV-1 capsid protein that forms a non-canonical pentamer. This non-canonical arrangement leads to an unusual capsid formation with a different shape distribution than normal and an unusual octahedral geometry which results in a cross-shaped hole in the capsids along with other unique structural properties. The structural analysis is well-designed and well-explained. There are only a couple of minor issues to be addressed before the manuscript is suitable for publication:
Response: We thank the Reviewer for being supportive of our study and for finding the work interesting.
1. The authors perform a nice structural comparison between the R18L capsid and previously-characterized R18A and R18G capsids. The R18 mutant capsids share some properties, but there are structural differences, particularly in terms of the distribution of oligomeric assemblies observed. The R18A and R18G have been characterized more thoroughly, so one additional thing that would tie the results together nicely would be to perform some characterization of R18L along the lines of what has been previously done for R18A and R18G. For example, the R18A and R18G mutants exhibit defective HIV particle formation and produce non-infectious particles. In addition, R18G exhibits enhanced stability in the absence of DTT relative to wild-type. Inclusion of these kinds or similar in vitro data would help tie the observed structural differences to observed biochemical differences.
Response: We have generated R18L virions, using the NL4-3 background, and analyzed 50 particles by tomography. We found only one virus particle to have assembled mature capsids - there were two of them, and the genome was encapsidated in one. This is consistent with observations from R18A, which indicated that many particles had multiple capsids (ref. 33 in revised ms). Unfortunately, neither the R18A study (ref. 33) nor the R18G study (ref. 34) analyzed the actual proportion of "apparently normal" cones. On the basis of our preliminary analysis of R18L, we believe that these are a very small minority. A substantially more thorough comparison of the different mutations will have to be performed in the future.
2. Check the references. Reference 16 on line 68 should actually be reference 15. I didn't check all of them, but if I found one, there might be other instances where something is mis-referenced.
Response: We thank the Reviewer for noticing the error. Refs. 15 and 16 (now 16 and 17 in the revised ms) are called correctly in the text, but were inadvertently switched in the Reference section. This has been corrected. We also checked all the other citations.
Reviewer 2 Report
Comments and Suggestions for Authors
The HIV capsid (as for other retroviruses as well) is able to assemble into multiple forms driven by subtle changes in the interfaces between subunits. Schirra et al. describes mutants in the capsid protein with greater variation in assembly as assessed by cryoEM, including a novel octahedral form. The manuscript is well written and is suitable for publication.
However, they only touch on the functional relevance of these mutants in the last paragraph where it is stated that they are non-infectious. The question is why? Also, are there other functional issues that differ from the wild type capsids? It would benefit the manuscript to clarify why these mutants are important.
Line 36:
"Capsid multifunctionality is underscored"
How? A bit of clarification how it relates to capsid structure would be useful.
Line 217:
Symmetrization will almost always give a higher resolution map, regardless of whether the symmetry is real. Why not assess the four-fold symmetry quantitatively from the C1 map?
Line 266;
Again, symmetry in the C1 map can be quantified.
Line 301:
"mixed symmetry capsids"
I'm not sure what this means. The issue of local symmetry is well justified. Globally, polyhedra can be closed (when all valencies are satisfied) or open with partial local symmetries. Even the typical HIV cone capsid as a closed polyhedron does not conform to a single well-defined symmetry. If a particle has a well-defined symmetry, it cannot be characterized as "mixed". If it has different local symmetries (even down to the different polygonal faces), one could call that "mixed". So I find the term poorly chosen.
Line 308:
I don't know what would make a possibility "formal". What is an "informal" possibility?
Line 319:
"icosahedral geometries of a fullerene cone"
This is not correct. The end cap on a cone may have partial icosahedral symmetry, but not necessarily. Better to describe it in terms of 7 and 5 pentagons on the two ends of the polyhedron.
Line 351:
"alternative assemblies"
Such alternatives are well known for other retroviruses where the capsids have even more variation. Why not reference them here.
Author Response
The HIV capsid (as for other retroviruses as well) is able to assemble into multiple forms driven by subtle changes in the interfaces between subunits. Schirra et al. describes mutants in the capsid protein with greater variation in assembly as assessed by cryoEM, including a novel octahedral form. The manuscript is well written and is suitable for publication.
However, they only touch on the functional relevance of these mutants in the last paragraph where it is stated that they are non-infectious. The question is why? Also, are there other functional issues that differ from the wild type capsids? It would benefit the manuscript to clarify why these mutants are important.
Response: We thank the Reviewer for being supportive of publication of our manuscript. The final point, on the functional relevance of the mutants, is well taken, and our paper is meant to provide the structural foundations for future studies. Previous studies of Arg18 mutants have concluded that the mutations cause defects in capsid architecture, and our work is in agreement with this. Importantly, we extend these findings by showing in molecular detail that Arg18 mutations cause a specific kind of defect: an alternative pentamer form that can incorporate in the capsid lattice as a single-pentamer or tetramer-of-pentamer vertex. The importance of this finding is emphasized in the title of the manuscript. We believe that further studies of these mutants is warranted, in order to better understand whether and how the capacity of HIV-1 CA for non-canonical assembly has biological relevance.
Line 36:
"Capsid multifunctionality is underscored"
How? A bit of clarification how it relates to capsid structure would be useful.
Response: We added the following sentence to the first paragraph: "These various capsid functions are mediated by interactions between the capsid and a growing list of host factors that bind to different sites displayed on the capsid surface." (lines 36-38 in the revised ms)
Line 217:
Symmetrization will almost always give a higher resolution map, regardless of whether the symmetry is real. Why not assess the four-fold symmetry quantitatively from the C1 map?
Response: We have now solved many structures (published and unpublished) centered on symmetry elements on the capsid surface. On the basis of those structures, we disagree with the Reviewer's assertion that symmetrization gives a higher resolution map regardless of whether the symmetry is real. However, a detailed discussion of this point is beyond the scope of the ms, so we have done as the Reviewer requested and assessed the four-fold symmetry quantitatively, using the self-rotation function. This analysis is now shown in the new Supplementary Figure 6.
Line 266;
Again, symmetry in the C1 map can be quantified.
Response: Please see above and Supplementary Figure 6.
Line 301:
"mixed symmetry capsids"
I'm not sure what this means. The issue of local symmetry is well justified. Globally, polyhedra can be closed (when all valencies are satisfied) or open with partial local symmetries. Even the typical HIV cone capsid as a closed polyhedron does not conform to a single well-defined symmetry. If a particle has a well-defined symmetry, it cannot be characterized as "mixed". If it has different local symmetries (even down to the different polygonal faces), one could call that "mixed". So I find the term poorly chosen.
Response: We use the term to indicated "mixtures of local symmetries" - which we agree is poorly chosen, so we no longer use it in the text.
Line 308:
I don't know what would make a possibility "formal". What is an "informal" possibility?
Response: This is old school formalistic science writing, and not necessary to make the point, so we removed it.
Line 319:
"icosahedral geometries of a fullerene cone"
This is not correct. The end cap on a cone may have partial icosahedral symmetry, but not necessarily. Better to describe it in terms of 7 and 5 pentagons on the two ends of the polyhedron.
Response: We agree that the description is confusing, related to the "mixed symmetry" issue above, and have re-written this and related section.
Line 351:
"alternative assemblies"
Such alternatives are well known for other retroviruses where the capsids have even more variation. Why not reference them here.
Response: We are not aware of any other retroviral CA protein that assembles into two different kinds of pentamers, let alone a tetramer-of-pentamers.
Reviewer 3 Report
Comments and Suggestions for Authors
This work provides a detailed description of the CA assemblied formed by Arg18 mutants (L,G,A) for HIV-1 capsid assembly. The work is clearly presented and enjoyable to read.
My only comments would be to perhaps explain within the introduction the biological function of the R18 pore and that even though the L G A mutants are not seen in vivo, the details of the study are still warranted because of the reasons the authors mention within the discussion. The ability to nanoengineer delivery machines is a worthwhile exploration in itself. Additionally references and reflections on other viruses are important. Perhaps this aspect can be fleshed out a bit more within the discussion.
Author Response
This work provides a detailed description of the CA assemblied formed by Arg18 mutants (L,G,A) for HIV-1 capsid assembly. The work is clearly presented and enjoyable to read.
My only comments would be to perhaps explain within the introduction the biological function of the R18 pore and that even though the L G A mutants are not seen in vivo, the details of the study are still warranted because of the reasons the authors mention within the discussion. The ability to nanoengineer delivery machines is a worthwhile exploration in itself. Additionally references and reflections on other viruses are important. Perhaps this aspect can be fleshed out a bit more within the discussion.
Response: We thank the Reviewer for being supportive of our publication. Our studies are consistent with previous studies indicating that the principal defect caused by Arg18 mutations manifests during the maturation step; that is, loss of infectivity is primarily due to loss of the ability to properly assemble a capsid. The Reviewer is correct, of course, that Arg18 has additional functions post-maturation (e.g., dNTP uptake), and as suggested we have rewritten the Discussion section to flesh out a bit more how these different Arg18 functions relate to each other.